# ADAPTIVE OPTIMIZERS WITH SPARSE GROUP LASSO

## ABSTRACT

We develop a novel framework that adds the regularizers to a family of adaptive optimizers in deep learning, such as MOMENTUM, ADAGRAD, ADAM, AMS-GRAD, ADAHESSIAN, and create a new class of optimizers, which are named GROUP MOMENTUM, GROUP ADAGRAD, GROUP ADAM, GROUP AMSGRAD and GROUP ADAHESSIAN, etc., accordingly. We establish theoretically proven convergence guarantees in the stochastic convex settings, based on primal-dual methods. We evaluate the regularized effect of our new optimizers on three large-scale real-world ad click datasets with state-of-the-art deep learning models. The experimental results reveal that compared with the original optimizers with the post-processing procedure which use the magnitude pruning method, the performance of the models can be significantly improved on the same sparsity level. Furthermore, in comparison to the cases without magnitude pruning, our methods can achieve extremely high sparsity with significantly better or highly competitive performance.

## 1 INTRODUCTION

With the development of deep learning, deep neural network (DNN) models have been widely used in various machine learning scenarios such as search, recommendation and advertisement, and achieved significant improvements. In the last decades, different kinds of optimization methods based on the variations of stochastic gradient descent (SGD) have been invented for training DNN models. However, most optimizers cannot directly produce sparsity which has been proven effective and efficient for saving computational resource and improving model performance especially in the scenarios of very high-dimensional data. Meanwhile, the simple rounding approach is very unreliable due to the inherent low accuracy of these optimizers.

In this paper, we develop a new class of optimization methods, that adds the regularizers especially sparse group lasso to prevalent adaptive optimizers, and retains the characteristics of the respective optimizers. Compared with the original optimizers with the post-processing procedure which use the magnitude pruning method, the performance of the models can be significantly improved on the same sparsity level. Furthermore, in comparison to the cases without magnitude pruning, the new optimizers can achieve extremely high sparsity with significantly better or highly competitive performance. In this section, we describe the two types of optimization methods, and explain the motivation of our work.

### 1.1 ADAPTIVE OPTIMIZATION METHODS

Due to the simplicity and effectiveness, adaptive optimization methods (Robbins & Monro, 1951; Polyak, 1964; Duchi et al., 2011; Zeiler, 2012; Kingma & Ba, 2015; Reddi et al., 2018; Yao et al., 2020) have become the de-facto algorithms used in deep learning. There are multiple variants, but they can be represented using the general update formula (Reddi et al., 2018):

$$x_{t+1} = x_t - \alpha_t m_t / \sqrt{V_t}, \tag{1}$$

where $\alpha_t$ is the step size, $m_t$ is the first moment term which is the weighted average of gradient $g_t$ and $V_t$ is the so called second moment term that adjusts updated velocity of variable $x_t$ in each direction. Here, $\sqrt{V_t} := V_t^{1/2}$, $m_t / \sqrt{V_t} := \sqrt{V_t}^{-1} \cdot m_t$. By setting different $m_t$, $V_t$ and $\alpha_t$, we can derive different adaptive optimizers including MOMENTUM (Polyak, 1964), ADAGRAD (Duchi et al., 2011), ADAM (Kingma & Ba, 2015), AMSGRAD (Reddi et al., 2018) and ADAHESSIAN (Yao et al., 2020), etc. See Table 1.

Table 1: Adaptive optimizers with choosing different $m_t$, $V_t$ and $\alpha_t$.

| Optimizer | $m_t$ | $V_t$ | $\alpha_t$ |
|---|---|---|---|
| SGD | $g_t$ | $\mathbb{I}$ | $\frac{\alpha}{\sqrt{t}}$ |
| MOMENTUM | $\gamma m_{t-1} + g_t$ | $\mathbb{I}$ | $\alpha$ |
| ADAGRAD | $g_t$ | $\mathrm{diag}(\sum_{i=1}^t g_i^2)/t$ | $\frac{\alpha}{\sqrt{t}}$ |
| ADAM | $\beta_1 m_{t-1} + (1-\beta_1)g_t$ | $\beta_2 V_{t-1} + (1-\beta_2)\mathrm{diag}(g_t^2)$ | $\frac{\alpha\sqrt{1-\beta_2^t}}{1-\beta_1^t}$ |
| AMSGRAD | $\beta_1 m_{t-1} + (1-\beta_1)g_t$ | $\max(V_{t-1}, \beta_2 V_{t-1} + (1-\beta_2)\mathrm{diag}(g_t^2))$ | $\frac{\alpha\sqrt{1-\beta_2^t}}{1-\beta_1^t}$ |
| ADAHESSIAN | $\beta_1 m_{t-1} + (1-\beta_1)g_t$ | $\beta_2 V_{t-1} + (1-\beta_2)D_t^2$ [*] | $\frac{\alpha\sqrt{1-\beta_2^t}}{1-\beta_1^t}$ |

[*] $D_t = \mathrm{diag}(H_t)$, where $H_t$ is the Hessian matrix.

## 1.2 REGULARIZED OPTIMIZATION METHODS

Follow-the-regularized-leader (FTRL) (McMahan & Streeter, 2010; McMahan et al., 2013) has been widely used in click-through rates (CTR) prediction problems, which adds $\ell_1$-regularization (lasso) to logistic regression and can effectively balance the performance of the model and the sparsity of features. The update formula (McMahan et al., 2013) is:

$$x_{t+1} = \arg\min_x g_{1:t} \cdot x + \frac{1}{2}\sum_{s=1}^t \sigma_s\|x - x_s\|_2^2 + \lambda_1\|x\|_1, \tag{2}$$

where $g_{1:t} = \sum_{s=1}^t g_s$, $\frac{1}{2}\sum_{s=1}^t \sigma_s\|x - x_s\|_2^2$ is the strong convex term that stabilizes the algorithm and $\lambda_1\|x\|_1$ is the regularization term that produces sparsity. However, it doesn't work well in DNN models since one input feature can correspond to multiple weights and lasso only can make single weight zero hence can't effectively delete zeros features.

To solve above problem, Ni et al. (2019) adds the $\ell_{21}$-regularization (group lasso) to FTRL, which is named G-FTRL. Yang et al. (2010) conducts the research on a group lasso method for online learning that adds $\ell_{21}$-regularization to the algorithm of Dual Averaging (DA) (Nesterov, 2009), which is named DA-GL. Even so, these two methods cannot been applied to other optimizers. Different scenarios are suitable for different optimizers in the deep learning fields. For example, MOMENTUM (Polyak, 1964) is typically used in computer vision; ADAM (Kingma & Ba, 2015) is used for training transformer models for natural language processing; and ADAGRAD (Duchi et al., 2011) is used for recommendation systems. If we want to produce sparsity of the model in some scenario, we have to change optimizer which probably influence the performance of the model.

## 1.3 MOTIVATION

Eq. (1) can be rewritten into this form:

$$x_{t+1} = \arg\min_x m_t \cdot x + \frac{1}{2\alpha_t}\|\sqrt{V_t}^{\frac{1}{2}}(x - x_t)\|_2^2. \tag{3}$$

Furthermore, we can rewrite Eq. (3) into

$$x_{t+1} = \arg\min_x m_{1:t} \cdot x + \sum_{s=1}^t \frac{1}{2\alpha_s}\|Q_s^{\frac{1}{2}}(x - x_s)\|_2^2, \tag{4}$$

where $m_{1:t} = \sum_{s=1}^t m_s$, $\sum_{s=1}^t Q_s/\alpha_s = \sqrt{V_t}/\alpha_t$. It is easy to prove that Eq. (3) and Eq. (4) are equivalent using the method of induction. The matrices $Q_s$ can be interpreted as generalized learning rates. To our best knowledge, $V_t$ of Eq. (1) of all the adaptive optimization methods are diagonal for the computation simplicity. Therefore, we consider $Q_s$ as diagonal matrices throughout this paper.

We find that Eq. (4) is similar to Eq. (2) except for the regularization term. Therefore, we add the regularization term $\Psi(x)$ to Eq. (4), which is the sparse group lasso penalty also including $\ell_2$-

regularization that can diffuse weights of neural networks. The concrete formula is:

$$\Psi_t(x) = \sum_{g=1}^{G} \left( \lambda_1 \|x^g\|_1 + \lambda_{21} \sqrt{d_{x^g}} \|A_t^{\frac{1}{2}} x^g\|_2 \right) + \lambda_2 \|x\|_2^2, \tag{5}$$

where $\lambda_1$, $\lambda_{21}$, $\lambda_2$ are regularization parameters of $\ell_1$, $\ell_{21}$, $\ell_2$ respectively, $G$ is the total number of groups of weights, $x^g$ is the weights of group $g$ and $d_{x^g}$ is the size of group $g$. In DNN models, each group is defined as the set of outgoing weights from a unit which can be an input feature, or a hidden neuron, or a bias unit (see, e.g., Scardapane et al. (2016)). $A_t$ can be arbitrary positive matrix satisfying $A_{t+1} \succeq A_t$, e.g., $A_t = \mathbb{I}$. In Section 2.1, we let $A_t = (\sum_{s=1}^{t} \frac{Q_s^g}{2\alpha_s} + \lambda_2 \mathbb{I})$ just for solving the closed-form solution directly, where $Q_s^g$ is a diagonal matrix whose diagonal elements are part of $Q_s$ corresponding to $x_g$. The ultimate update formula is:

$$x_{t+1} = \arg\min_x m_{1:t} \cdot x + \sum_{s=1}^{t} \frac{1}{2\alpha_s} \|Q_s^{\frac{1}{2}}(x - x_s)\|_2^2 + \Psi_t(x). \tag{6}$$

### 1.4 OUTLINE OF CONTENTS

The rest of the paper is organized as follows. In Section 1.5, we introduce the necessary notations and technical background.

In Section 2, we present the closed-form solution of Eq. (4) and the algorithm of general framework of adaptive optimization methods with sparse group lasso. We prove the algorithm is equivalent to adaptive optimization methods when regularization terms vanish. In the end, we give two concrete examples of the algorithm.[1]

In Section 3, we derive the regret bounds of the method and convergence rates.

In Section 4, we validate the performance of new optimizers in the public datasets.

In Section 5, we summarize the conclusion.

Appendices A-B list the details of GROUP ADAM and Group Adagrad respectively. Appendices C-F contain technical proofs of our main results and Appendix G includes the details of the empirical results of Section 4.4.

### 1.5 NOTATIONS AND TECHNICAL BACKGROUND

We use lowercase letters to denote scalars and vectors, and uppercase letters to denote matrices. We denote a sequence of vectors by subscripts, that is, $x_1, \ldots, x_t$, and entries of each vector by an additional subscript, e.g., $x_{t,i}$. We use the notation $g_{1:t}$ as a shorthand for $\sum_{s=1}^{t} g_s$. Similarly we write $m_{1:t}$ for a sum of the first moment $m_t$, and $f_{1:t}$ to denote the function $f_{1:t}(x) = \sum_{s=1}^{t} f_s(x)$. Let $M_t = [m_1 \cdots m_t]$ denote the matrix obtained by concatenating the vector sequence $\{m_t\}_{t \geq 1}$ and $M_{t,i}$ denote the $i$-th row of this matrix which amounts to the concatenation of the $i$-th component of each vector. The notation $A \succeq 0$ (resp. $A \succ 0$) for a matrix A means that A is symmetric and positive semidefinite (resp. definite). Similarly, the notations $A \succeq B$ and $A \succ B$ mean that $A - B \succeq 0$ and $A - B \succ 0$ respectively, and both tacitly assume that $A$ and $B$ are symmetric. Given $A \succeq 0$, we write $A^{\frac{1}{2}}$ for the square root of $A$, the unique $X \succeq 0$ such that $XX = A$ (McMahan & Streeter (2010), Section 1.4).

Let $\mathcal{E}$ be a finite-dimension real vector space, endowed with the Mahalanobis norm $\|\cdot\|_A$ which is denoted by $\|\cdot\|_A = \sqrt{\langle \cdot, A \cdot \rangle}$ as induced by $A \succ 0$. Let $\mathcal{E}^*$ be the vector space of all linear functions on $\mathcal{E}$. The dual space $\mathcal{E}^*$ is endowed with the dual norm $\|\cdot\|_A^* = \sqrt{\langle \cdot, A^{-1} \cdot \rangle}$.

Let $\mathcal{Q}$ be a closed convex set in $\mathcal{E}$. A continuous function $h(x)$ is called *strongly convex* on $\mathcal{Q}$ with norm $\|\cdot\|_H$ if $\mathcal{Q} \subseteq \text{dom } h$ and there exists a constant $\sigma > 0$ such that for all $x, y \in \mathcal{Q}$ and $\alpha \in [0, 1]$ we have

$$h(\alpha x + (1 - \alpha)y) \leq \alpha h(x) + (1 - \alpha)h(y) - \frac{1}{2}\sigma\alpha(1 - \alpha)\|x - y\|_H^2.$$

---

[1]To fulfill research interest of optimization methods, we will release the code in the future.

The constant $\sigma$ is called the *convexity parameter* of $h(x)$, or the *modulus* of strong convexity. We also denote by $\|\cdot\|_h = \|\cdot\|_H$. Further, if $h$ is differential, we have

$$h(y) \geq h(x) + \langle \nabla h(x), y - x \rangle + \frac{\sigma}{2}\|x - y\|_h^2.$$

We use online convex optimization as our analysis framework. On each round $t = 1, \ldots, T$, a convex loss function $f_t : \mathcal{Q} \mapsto \mathbb{R}$ is chosen, and we pick a point $x_t \in \mathcal{Q}$ hence get loss $f_t(x_t)$. Our goal is minimizing the *regret* which is defined as the quantity

$$\mathcal{R}_T = \sum_{t=1}^T f_t(x_t) - \min_{x \in \mathcal{Q}} \sum_{t=1}^T f_t(x). \tag{7}$$

Online convex optimization can be seen as a generalization of stochastic convex optimization. Any regret minimizing algorithm can be converted to a stochastic optimization algorithm with convergence rate $O(\mathcal{R}_T/T)$ using an online-to-batch conversion technique (Littlestone, 1989).

In this paper, we assume $\mathcal{Q} \equiv \mathcal{E} = \mathbb{R}^n$, hence we have $\mathcal{E}^* = \mathbb{R}^n$. We write $s^T x$ or $s \cdot x$ for the standard inner product between $s, x \in \mathbb{R}^n$. For the standard Euclidean norm, $\|x\| = \|x\|_2 = \sqrt{\langle x, x \rangle}$ and $\|s\|_* = \|s\|_2$. We also use $\|x\|_1 = \sum_{i=1}^n |x^{(i)}|$ and $\|x\|_\infty = \max_i |x^{(i)}|$ to denote $\ell_1$-norm and $\ell_\infty$-norm respectively, where $x^{(i)}$ is the $i$-th element of $x$.

## 2 ALGORITHM

### 2.1 CLOSED-FORM SOLUTION

We will derive the closed-form solution of Eq. (6) with specific $A_t$ and Algorithm 1 with slight modification in this section. We have the following theorem.

**Theorem 1.** *Given $A_t = (\sum_{s=1}^t \frac{Q_s^g}{2\alpha_s} + \lambda_2\mathbb{I})$ of Eq. (5), $z_t = z_{t-1} + m_t - \frac{Q_t}{\alpha_t}x_t$ at each iteration $t = 1, \ldots, T$ and $z_0 = \mathbf{0}$, the optimal solution of Eq. (6) is updated accordingly as follows:*

$$x_{t+1} = (\sum_{s=1}^t \frac{Q_s}{\alpha_s} + 2\lambda_2\mathbb{I})^{-1} \max(1 - \frac{\sqrt{d_{x_t^g}}\lambda_{21}}{\|\tilde{s}_t\|_2}, 0)s_t \tag{8}$$

*where the $i$-th element of $s_t$ is defined as*

$$s_{t,i} = \begin{cases} 0 & \text{if } |z_{t,i}| \leq \lambda_1, \\ \text{sign}(z_{t,i})\lambda_1 - z_{t,i} & \text{otherwise,} \end{cases} \tag{9}$$

$\tilde{s}_t$ *is defined as*

$$\tilde{s}_t = (\sum_{s=1}^t \frac{Q_s}{2\alpha_s} + \lambda_2\mathbb{I})^{-1}s_t \tag{10}$$

*and $\sum_{s=1}^t \frac{Q_s}{\alpha_s}$ is the diagonal and positive definite matrix.*

The proof of Theorem 1 is given in Appendix C. We slightly modify (8) where we let $\tilde{s}_t = s_t$. Our purpose is to let every entry of the group have the same effect of $\ell_{21}$-regularization. Hence, we get Algorithm 1. Furthermore, we have the following theorem which shows the relationship between Algorithm 1 and adaptive optimization methods. The proof is given in Appendix D.

**Theorem 2.** *If regularization terms of Algorithm 1 vanish, Algorithm 1 is equivalent to Eq. (1).*

### 2.2 CONCRETE EXAMPLES

Using Algorithm 1, we can easily derive the new optimizers based on ADAM (Kingma & Ba, 2015), ADAGRAD (Duchi et al., 2011) which we call GROUP ADAM, GROUP ADAGRAD respectively.

GROUP ADAM

The detail of the algorithm is given in Appendix A. From Theorem 2, we know that when $\lambda_1$, $\lambda_2$, $\lambda_{21}$ are all zeros, Algorithm 2 is equivalent to ADAM (Kingma & Ba, 2015).

---

**Algorithm 1** Generic framework of adaptive optimization methods with sparse group lasso

1: **Input:** parameters $\lambda_1, \lambda_{21}, \lambda_2$
   $x_1 \in \mathbb{R}^n$, step size $\{\alpha_t > 0\}_{t=1}^T$, sequence of functions $\{\phi_t, \psi_t\}_{t=1}^T$, initialize $z_0 = \mathbf{0}, V_0 = \mathbf{0}, \alpha_0 = \mathbf{0}$
2: **for** $t = 1$ **to** $T$ **do**
3:     $g_t = \nabla f_t(x_t)$
4:     $m_t = \phi_t(g_1, \ldots, g_t)$ and $V_t = \psi_t(g_1, \ldots, g_t)$
5:     $\frac{Q_t}{\alpha_t} = \frac{\sqrt{V_t}}{\alpha_t} - \frac{\sqrt{V_{t-1}}}{\alpha_{t-1}}$
6:     $z_t \leftarrow z_{t-1} + m_t - \frac{Q_t}{\alpha_t} x_t$
7:     **for** $i \in \{1, \ldots, n\}$ **do**
8:         $s_{t,i} = \begin{cases} 0 & \text{if } |z_{t,i}| \leq \lambda_1 \\ \text{sign}(z_{t,i})\lambda_1 - z_{t,i} & \text{otherwise.} \end{cases}$
9:     **end for**
10:    $x_{t+1} = (\frac{\sqrt{V_t}}{\alpha_t} + 2\lambda_2 \mathbb{I})^{-1} \max(1 - \frac{\sqrt{d_{x_t^g}}\lambda_{21}}{\|s_t\|_2}, 0) s_t$
11: **end for**

---

### GROUP ADAGRAD

The detail of the algorithm is given in Appendix B. Similarly, from Theorem 2, when $\lambda_1, \lambda_2, \lambda_{21}$ are all zeros, Algorithm 3 is equivalent to ADAGRAD (Duchi et al., 2011). Furthermore, we can find that when $\lambda_{21} = 0$, Algorithm 3 is equivalent to FTRL (McMahan et al., 2013). Therefore, GROUP ADAGRAD can also be called GROUP FTRL from the research of Ni et al. (2019).

Similarly, GROUP MOMENTUM, GROUP AMSGRAD, GROUP ADAHESSIAN, etc., can be derived from MOMENTUM (Polyak, 1964), AMSGRAD (Reddi et al., 2018), ADAHESSIAN (Yao et al., 2020), etc., with the same framework and we will not list the details.

## 3 CONVERGENCE AND REGRET ANALYSIS

Using the framework developed in Nesterov (2009); Xiao (2010); Duchi et al. (2011), we have the following theorem providing the bound of the regret.

**Theorem 3.** *Let the sequence $\{x_t\}$ be defined by the update* (6) *and*

$$x_1 = \arg\min_{x \in \mathcal{Q}} \frac{1}{2}\|x - c\|_2^2, \tag{11}$$

*where $c$ is an arbitrary constant vector. Suppose $f_t(x)$ is convex for any $t \geq 1$ and there exists an optimal solution $x^*$ of $\sum_{t=1}^T f_t(x)$, i.e., $x^* = \arg\min_{x \in \mathcal{Q}} \sum_{t=1}^T f_t(x)$, which satisfies the condition*

$$\langle m_{t-1}, x_t - x^* \rangle \geq 0, \quad t \in [T], \tag{12}$$

*where $m_t$ is the weighted average of the gradient $f_t(x_t)$ and $[T] = \{1, \ldots, T\}$ for simplicity. Without loss of generality, we assume*

$$m_t = \gamma m_{t-1} + g_t, \tag{13}$$

*where $\gamma < 1$ and $m_0 = 0$. Then*

$$\mathcal{R}_T \leq \Psi_T(x^*) + \sum_{t=1}^T \frac{1}{2\alpha_t}\|Q_t^{\frac{1}{2}}(x^* - x_t)\|_2^2 + \frac{1}{2}\sum_{t=1}^T \|m_t\|_{h_{t-1}^*}^2, \tag{14}$$

*where $\|\cdot\|_{h_t^*}$ is the dual norm of $\|\cdot\|_{h_t}$. $h_t$ is 1-strongly convex with respect to $\|\cdot\|_{\sqrt{V_t}/\alpha_t}$ for $t \in [T]$ and $h_0$ is 1-strongly convex with respect to $\|\cdot\|_2$.*

The proof of Theorem 3 is given in Appendix E. Since in most of adaptive optimizers, $V_t$ is the weighted average of $\text{diag}(g_t^2)$, without loss of generality, we assume $\alpha_t = \alpha$ and

$$V_t = \eta V_{t-1} + \text{diag}(g_t^2), \quad t \geq 1, \tag{15}$$

where $V_0 = 0$ and $\eta \leq 1$. Hence, we have the following lemma whose proof is given in Appendix F.1.

**Lemma 1.** *Suppose $V_t$ is the weighted average of the square of the gradient which is defined by (15), $\alpha_t = \alpha$, $m_t$ is defined by (13) and $V_t$ satisfies the following arbitrary conditions:*

1. *$\eta = 1$,*

2. *$\eta < 1$, $\eta \geq \gamma$ and $\kappa V_t \succeq V_{t-1}$ for all $t \geq 1$ where $\kappa < 1$.*

*Then we have*

$$\sum_{t=1}^{T} \|m_t\|^2_{(\frac{\sqrt{V_t}}{\alpha_t})^{-1}} < \frac{2\alpha}{1-\nu} \sum_{i=1}^{d} \|M_{T,i}\|_2, \tag{16}$$

*where $\nu = \max(\gamma, \kappa)$ and $d$ is the dimension of $x_t$.*

We can always add $\delta^2 \mathbb{I}$ to $V_t$ at each step to ensure $V_t \succ 0$. Therefore, $h_t(x)$ is 1-strongly convex with respect to $\| \cdot \|_{\sqrt{\delta^2 \mathbb{I} + V_t}/\alpha_t}$. Let $\delta \geq \max_{t \in [T]} \|g_t\|_\infty$, for $t > 1$, we have

$$\|m_t\|^2_{h^*_{t-1}} = \left\langle m_t, \alpha_t(\delta^2 \mathbb{I} + V_{t-1})^{-\frac{1}{2}} m_t \right\rangle \leq \left\langle m_t, \alpha_t \left(\text{diag}(g_t^2) + \eta V_{t-1}\right)^{-\frac{1}{2}} m_t \right\rangle$$
$$= \left\langle m_t, \alpha_t V_t^{-\frac{1}{2}} m_t \right\rangle = \|m_t\|^2_{(\frac{\sqrt{V_t}}{\alpha_t})^{-1}}. \tag{17}$$

For $t = 1$, we have

$$\|m_1\|^2_{h^*_0} = \left\langle m_1, \alpha_1(\delta^2 \mathbb{I} + \mathbb{I})^{-\frac{1}{2}} m_1 \right\rangle \leq \left\langle m_1, \alpha_1 \left(\text{diag}^{-\frac{1}{2}}(g_1^2)\right) m_1 \right\rangle$$
$$= \left\langle m_1, \alpha_1 V_1^{-\frac{1}{2}} m_1 \right\rangle = \|m_1\|^2_{(\frac{\sqrt{V_1}}{\alpha_1})^{-1}}. \tag{18}$$

From (17), (18) and Lemma 1, we have

**Lemma 2.** *Suppose $V_t$, $m_t$, $\alpha_t$, $\nu$, $d$ are defined the same as Lemma 1, $\max_{t \in [T]} \|g_t\|_\infty \leq \delta$, $\| \cdot \|^2_{h^*_t} = \left\langle \cdot, \alpha_t(\delta^2 \mathbb{I} + V_t)^{-\frac{1}{2}} \cdot \right\rangle$ for $t \geq 1$ and $\| \cdot \|^2_{h^*_0} = \left\langle \cdot, \alpha_1 \left((\delta^2 + 1)\mathbb{I}\right)^{-\frac{1}{2}} \cdot \right\rangle$. Then*

$$\sum_{t=1}^{T} \|m_t\|^2_{h^*_{t-1}} < \frac{2\alpha}{1-\nu} \sum_{i=1}^{d} \|M_{T,i}\|_2. \tag{19}$$

Therefore, from Theorem 3 and Lemma 2, we have

**Corollary 1.** *Suppose $V_t$, $m_t$, $\alpha_t$, $h^*_t$, $\nu$, $d$ are defined the same as Lemma 2, there exist constants $G$, $D_1$, $D_2$ such that $\max_{t \in [T]} \|g_t\|_\infty \leq G \leq \delta$, $\|x^*\|_\infty \leq D_1$ and $\max_{t \in [T]} \|x_t - x^*\|_\infty \leq D_2$. Then*

$$\mathcal{R}_T < dD_1 \left(\lambda_1 + \lambda_{21}(\frac{\sqrt{T}G}{2\alpha} + \lambda_2)^{\frac{1}{2}} + \lambda_2 D_1\right) + dG \left(\frac{D_2^2}{2\alpha} + \frac{\alpha}{(1-\nu)^2}\right) \sqrt{T}. \tag{20}$$

The proof of Corollary 1 is given in F.2. Furthermore, from Corollary 1, we have

**Corollary 2.** *Suppose $m_t$ is defined as (13), $\alpha_t = \alpha$ and satisfies the condition (19). There exist constants $G$, $D_1$, $D_2$ such that $tG^2 \mathbb{I} \succeq V_t$, $\max_{t \in [T]} \|g_t\|_\infty \leq G$, $\|x^*\|_\infty \leq D_1$ and $\max_{t \in [T]} \|x_t - x^*\|_\infty \leq D_2$. Then*

$$\mathcal{R}_T < dD_1 \left(\lambda_1 + \lambda_{21}(\frac{\sqrt{T}G}{2\alpha} + \lambda_2)^{\frac{1}{2}} + \lambda_2 D_1\right) + dG \left(\frac{D_2^2}{2\alpha} + \frac{\alpha}{(1-\nu)^2}\right) \sqrt{T}. \tag{21}$$

Therefore, we know that the regret of the update (6) is $O(\sqrt{T})$ and can achieve the optimal convergence rate $O(1/\sqrt{T})$ under the conditions of Corollary 1 or Corollary 2.

## 4 EXPERIMENTS

### 4.1 EXPERIMENT SETUP

We test the algorithms on three different large-scale real-world datasets with different neural network structures. These datasets are various display ads logs for the purpose of predicting ads CTR. The details are as follows.

a) The Avazu CTR dataset (Avazu, 2015) contains approximately 40M samples and 22 categorical features over 10 days. In order to handle categorical data, we use the one-hot-encoding based embedding technique (see, e.g., Wang et al. (2017), Section 2.1 or Naumov et al. (2019), Section 2.1.1) and get 9.4M features in total. For this dataset, the samples from the first 9 days (containing 8.7M one-hot features) are used for training, while the rest is for testing. Our DNN model follows the basic structure of most deep CTR models. Specifically, the model comprises one embedding layer, which maps each one-hot feature into 16-dimensional embeddings, and four fully connected layers (with output dimension of 64, 32, 16 and 1, respectively) in sequence.

b) The iPinYou dataset[2] (iPinYou, 2013) is another real-world dataset for ad click logs over 21 days. The dataset contains 16 categorical features[3]. After one-hot encoding, we get a dataset containing 19.5M instances with 1033.1K input dimensions. We keep the original train/test splitting scheme, where the training set contains 15.4M samples with 937.7K one-hot features. We use Outer Product-based Neural Network (OPNN) (Qu et al., 2016), and follow the standard settings of Qu et al. (2016), i.e., one embedding layer with the embedding dimension of 10, one product layer and three hidden layers of size 512, 256, 128 respectively where we set dropout rate at 0.5.

c) The third dataset is the Criteo Display Ads dataset (Criteo, 2014) which contains approximately 46M samples over 7 days. There are 13 integer features and 26 categorical features. After one-hot encoding of categorical features, we have total 33.8M features. We split the dataset into 7 partitions in chronological order and select the earliest 6 parts for training which contains 29.6M features and the rest for testing though the dataset has no timestamp. We use Deep & Cross Network (DCN) (Wang et al., 2017) and choose the following settings[4]: one embedding layer with embedding dimension 8, two deep layers of size 64 each, and two cross layers.

For the convenience of discussion, we use MLP, OPNN and DCN to represent the aforementioned three datasets coupled with their corresponding models. It is obvious that the embedding layer has most of parameters of the neural networks when the features have very high dimension, therefore we just add the regularization terms to the embedding layer. Furthermore, each embedding vector is considered as a group, and a visual comparison between $\ell_1$, $\ell_{21}$ and mixed regularization effect is given in Fig. 2 of Scardapane et al. (2016).

We treat the training set as the streaming data, hence we train 1 epoch with a batch size of 512 and do the validation. The experiments are conducted with 4-9 workers and 2-3 parameter servers, which depends on the different sizes of the datasets. We use the area under the receiver-operator curve (AUC) as the evaluation criterion since it is widely used in evaluating classification problems. Besides, some work validates AUC as a good measurement in CTR estimation (Graepel et al., 2010). We explore 5 learning rates from 1e-5 to 1e-1 with increments of 10x and choose the one with the best AUC for each new optimizer in the case of no regularization terms (It is equivalent to the original optimizer according to Theorem 2). All the experiments are run 5 times repeatedly and tested statistical significance using t-test. Without loss of generality, we choose two new optimizers to validate the performance, which are GROUP ADAM and GROUP ADAGRAD.

### 4.2 ADAM VS. GROUP ADAM

First, we compare the performance of the two optimizers on the same sparsity level. We keep $\lambda_1$, $\lambda_2$ be zeros and choose different values of $\lambda_{21}$ of Algorithm 2, i.e., GROUP ADAM, and achieve the

---

[2] We only use the data from season 2 and 3 because of the same data schema.

[3] See https://github.com/Atomu2014/Ads-RecSys-Datasets/ for details.

[4] Limited by training resources available, we don't use the optimal hyperparameter settings of Wang et al. (2017).

same sparsity with ADAM that uses the magnitude pruning method, i.e., sort the norm of embedding vector from largest to smallest, and keep top N embedding vectors which depend on the sparsity when finish the training. Table 2 reports the average results of the two optimizers in the three datasets. Note that GROUP ADAM significantly outperforms ADAM on the AUC metric on the same sparsity level for most experiments. Furthermore, as shown in Figure 1, the same $\ell_{21}$-regularization strength $\lambda_{21}$ has different effects of sparsity and accuracy on different datasets. The best choice of $\lambda_{21}$ depends on the dataset as well as the application (For example, if the memory of serving resource is limited, sparsity might be relative more important). One can trade off accuracy to get more sparsity by increasing the value of $\lambda_{21}$.

Table 2: AUC for the two optimizers and sparsity (feature rate) in parentheses. The best AUC for each dataset on each sparsity level is bolded. The p-value of the t-test of AUC is also listed.

| $\lambda_{21}$ | **MLP** | | | **OPNN** | | | **DCN** | | |
| GROUP ADAM | ADAM | GROUP ADAM | P-Value | ADAM | GROUP ADAM | P-Value | ADAM | GROUP ADAM | P-Value |
|---|---|---|---|---|---|---|---|---|---|
| 1e-4 | 0.7452 (0.974) | **0.7461** (0.974) | 0.025 | 0.7551 (0.078) | **0.7595** (0.078) | 0.086 | 0.8018 (0.518) | **0.8022** (0.518) | 0.105 |
| 5e-4 | 0.7464 (0.864) | **0.7468** (0.864) | 0.466 | 0.7491 (0.039) | **0.7573** (0.039) | 0.091 | 0.8017 (0.062) | **0.8019** (0.062) | 0.487 |
| 1e-3 | 0.7452 (0.701) | **0.7468** (0.701) | 0.058 | 0.7465 (0.032) | **0.7595** (0.032) | 0.014 | 0.8017 (0.018) | 0.8017 (0.018) | 0.943 |
| 5e-3 | 0.7452 (0.132) | **0.7464** (0.132) | 0.155 | 0.7509 (0.018) | **0.7561** (0.018) | 0.041 | 0.7995 (4.2e-3) | **0.8007** (4.2e-3) | 9.11e-3 |
| 1e-2 | 0.7430 (0.038) | **0.7466** (0.038) | 3.73e-4 | 0.7396 (9.2e-3) | **0.7493** (9.2e-3) | 0.031 | 0.7972 (2.5e-3) | **0.7999** (2.5e-3) | 5.97e-7 |

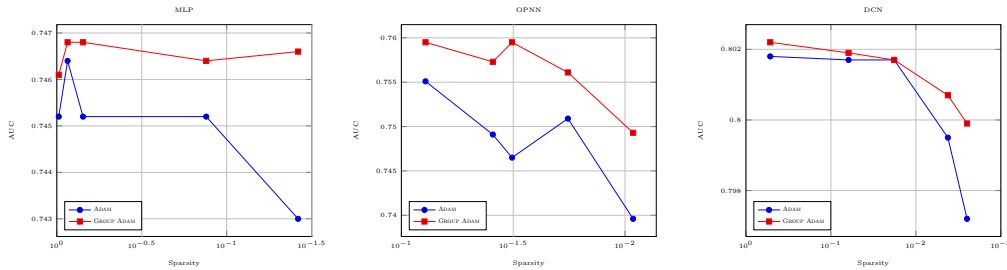

Figure 1: The AUC across different sparsity on two optimizers for the three datasets. The x-axis is sparsity (number of non-zero features whose embedding vectors are not equal to **0** divided by the total number of features present in the training data). The y-axis is AUC.

Next, we compare the performance of ADAM without post-processing procedure, i.e., no magnitude pruning, and GROUP ADAM with appropriate regularization terms which we choose in Table 3 on the AUC metric. In general, good default settings of $\lambda_2$ is 1e-5. The results are shown in Table 4. Note that compared with ADAM, GROUP ADAM with appropriate regularization terms can achieve significantly better or highly competitive performance with producing extremely high sparsity.

### 4.3 ADAGRAD VS. GROUP ADAGRAD

We compare with the performance of ADAGRAD without magnitude pruning and GROUP ADAGRAD with appropriate regularization terms which we choose in Table 5 on the AUC metric. The results are shown in Table 6. Again note that in comparison to ADAGRAD, GROUP ADAGRAD can not only achieve significantly better or highly competitive performance of AUC, but also effectively and efficiently reduce the dimensions of the features.

Table 3: The regularization terms of GROUP ADAM of three datasets.

| Dataset | $\lambda_1$ | $\lambda_{21}$ | $\lambda_2$ |
|---|---|---|---|
| MLP | 5e-3 | 1e-2 | 1e-5 |
| OPNN | 8e-5 | 1e-5 | 1e-5 |
| DCN | 4e-4 | 5e-4 | 1e-5 |

Table 4: AUC for three datasets and sparsity (feature rate) in parentheses. The best value for each dataset is bolded. The p-value of t-test is also listed.

| Dataset | ADAM | GROUP ADAM | P-Value |
|---|---|---|---|
| MLP | 0.7458 (1.000) | **0.7486 (0.018)** | 1.10e-3 (2.69e-11) |
| OPNN | 0.7588 (0.827) | **0.7617 (0.130)** | 0.289 (6.20e-11) |
| DCN | **0.8021** (1.000) | 0.8019 (**0.030**) | 0.422 (1.44e-11) |

Table 5: The regularization terms of GROUP ADAGRAD of three datasets.

| Dataset | $\lambda_1$ | $\lambda_{21}$ | $\lambda_2$ |
|---|---|---|---|
| MLP | 0 | 1e-2 | 1e-5 |
| OPNN | 8e-5 | 8e-5 | 1e-5 |
| DCN | 0 | 4e-3 | 1e-5 |

Table 6: AUC for three datasets and sparsity (feature rate) in parentheses. The best value for each dataset is bolded. The p-value of t-test is also listed.

| Dataset | ADAGRAD | GROUP ADAGRAD | P-Value |
|---|---|---|---|
| MLP | 0.7453 (1.000) | **0.7469 (0.063)** | 0.106 (1.51e-9) |
| OPNN | 0.7556 (0.827) | **0.7595 (0.016)** | 0.026 ($< 2.2$e-16) |
| DCN | 0.7975 (1.000) | **0.7978 (0.040)** | 0.198 (3.94e-11) |

## 4.4 DISCUSSION

In this section we will discuss the hyperparameters of emdedding dimension, $\ell_1$-regularization and $\ell_{21}$-regularization to show how these hyperparameters affect the effects of regularization.

**Embedding Dimension** Table 7 of Appendix G reports the average results of different embedding dimensions of MLP, whose optimizer is GROUP ADAM and regularization terms are same to MLP of Table 5. Note that the sparsity increases with the growth of the embedding dimension. The reason is that the square root of the embedding dimension is the multiplier of $\ell_{21}$-regularization.

**$\ell_1$ vs. $\ell_{21}$** From lines 8 and 10 of Algorithm 1, we know that if $z_t$ has the same elements, the values of $\ell_1$ and $\ell_{21}$, i.e., $\lambda_1$ and $\lambda_{21}$, have the same regularization effects. However, this situation almost cannot be happen in reality. Without loss of generality, we set optimizer, $\lambda_2$ and embedding dimension be GROUP ADAM, 1e-5 and 16 respectively, and choose different values of $\lambda_1$, $\lambda_{21}$. The results on MLP are shown in Table 8 of Appendix G. It is obvious that $\ell_{21}$-regularization is much more effective than $\ell_1$-regularization in producing sparsity. For example, when $\lambda_1 = 0$ and $\lambda_{21} = $ 5e-3, the feature sparsity is 0.136, while for $\lambda_1 = $ 5e-3 and $\lambda_{21} = 0$, the feature sparsity is 0.470. Therefore, if just want to produce sparsity, we can only tune $\lambda_{21}$ and use default settings for $\lambda_2$ and $\lambda_1$, i.e., $\lambda_2 = $ 1e-5 and $\lambda_1 = 0$.

## 5 CONCLUSION

In this paper, we propose a novel framework that adds the regularization terms to a family of adaptive optimizers for producing sparsity of DNN models. We apply this framework to create a new class of optimizers. We provide closed-form solutions and algorithms with slight modification. We built the relation between new and original optimizers, i.e., our new optimizers become equivalent with the corresponding original ones, once the regularization terms vanish. We theoretically prove the convergence rate of the regret and also conduct empirical evaluation on the proposed optimizers in comparison to the original optimizers with and without magnitude pruning. The results clearly demonstrate the advantages of our proposed optimizers in both getting significantly better performance and producing sparsity. Finally, it would be interesting in the future to investigate the convergence in non-convex settings and evaluate our optimizers on more applications from fields such as compute vision, natural language processing and etc.

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

## APPENDIX

## A GROUP ADAM

---

**Algorithm 2** Group Adam

---

1: **Input:** parameters $\lambda_1, \lambda_{21}, \lambda_2, \beta_1, \beta_2, \epsilon$
   $x_1 \in \mathbb{R}^n$, step size $\alpha$, initialize $z_0 = \mathbf{0}, \hat{m}_0 = \mathbf{0}, \hat{V}_0 = \mathbf{0}, V_0 = \mathbf{0}$
2: **for** $t = 1$ **to** $T$ **do**
3:     $g_t = \nabla f_t(x_t)$
4:     $\hat{m}_t \leftarrow \beta_1 \hat{m}_{t-1} + (1 - \beta_1) g_t$
5:     $m_t = \hat{m}_t / (1 - \beta_1^t)$
6:     $\hat{V}_t \leftarrow \beta_2 \hat{V}_{t-1} + (1 - \beta_2) \mathrm{diag}(g_t^2)$
7:     $V_t = \hat{V}_t / (1 - \beta_2^t)$
8:     $Q_t = \begin{cases} \sqrt{V_t} - \sqrt{V_{t-1}} + \epsilon \mathbb{I} & t = 1 \\ \sqrt{V_t} - \sqrt{V_{t-1}} & t > 1 \end{cases}$
9:     $z_t \leftarrow z_{t-1} + m_t - \frac{1}{\alpha} Q_t x_t$
10:    **for** $i \in \{1, \ldots, n\}$ **do**
11:       $s_{t,i} = \begin{cases} 0 & \text{if } |z_{t,i}| \leq \lambda_1 \\ \mathrm{sign}(z_{t,i}) \lambda_1 - z_{t,i} & \text{otherwise.} \end{cases}$
12:    **end for**
13:     $x_{t+1} = (\frac{\sqrt{V_t} + \epsilon \mathbb{I}}{\alpha} + 2\lambda_2 \mathbb{I})^{-1} \max(1 - \frac{\sqrt{d_{x_t^g}} \lambda_{21}}{\|s_t\|_2}, 0) s_t$
14: **end for**

---

## B GROUP ADAGRAD

---

**Algorithm 3** Group Adagrad

---

1: **Input:** parameters $\lambda_1, \lambda_{21}, \lambda_2, \epsilon$
   $x_1 \in \mathbb{R}^n$, step size $\alpha$, initialize $z_0 = \mathbf{0}, V_0 = \mathbf{0}$
2: **for** $t = 1$ **to** $T$ **do**
3:     $g_t = \nabla f_t(x_t)$
4:     $m_t = g_t$
5:     $V_t = \begin{cases} V_{t-1} + \mathrm{diag}(g_t^2) + \epsilon \mathbb{I} & t = 1 \\ V_{t-1} + \mathrm{diag}(g_t^2) & t > 1 \end{cases}$
6:     $Q_t = \sqrt{V_t} - \sqrt{V_{t-1}}$
7:     $z_t \leftarrow z_{t-1} + m_t - \frac{1}{\alpha} Q_t x_t$
8:    **for** $i \in \{1, \ldots, n\}$ **do**
9:       $s_{t,i} = \begin{cases} 0 & \text{if } |z_{t,i}| \leq \lambda_1 \\ \mathrm{sign}(z_{t,i}) \lambda_1 - z_{t,i} & \text{otherwise.} \end{cases}$
10:    **end for**
11:     $x_{t+1} = (\frac{\sqrt{V_t}}{\alpha} + 2\lambda_2 \mathbb{I})^{-1} \max(1 - \frac{\sqrt{d_{x_t^g}} \lambda_{21}}{\|s_t\|_2}, 0) s_t$
12: **end for**

---

## C PROOF OF THEOREM 1

*Proof.*

$$
\begin{aligned}
x_{t+1} &= \arg\min_x m_{1:t} \cdot x + \sum_{s=1}^{t} \frac{1}{2\alpha_s} (x - x_s)^T Q_s (x - x_s) + \Psi_t(x) \\
&= \arg\min_x m_{1:t} \cdot x + \sum_{s=1}^{t} \frac{1}{2\alpha_s} (\|Q_s^{\frac{1}{2}} x\|_2^2 - 2x \cdot (Q_s x_s) + \|Q_s^{\frac{1}{2}} x_s\|_2^2) + \Psi_t(x) \quad\quad (22) \\
&= \arg\min_x \left( m_{1:t} - \sum_{s=1}^{t} \frac{Q_s}{\alpha_s} x_s \right) \cdot x + \sum_{s=1}^{t} \frac{1}{2\alpha_s} \|Q_s^{\frac{1}{2}} x\|_2^2 + \Psi_t(x).
\end{aligned}
$$

Define $z_{t-1} = m_{1:t-1} - \sum_{s=1}^{t-1} \frac{Q_s}{\alpha_s} x_s$ $(t \geq 2)$ and we can calculate $z_t$ as

$$z_t = z_{t-1} + m_t - \frac{Q_t}{\alpha_t} x_t, \quad t \geq 1. \tag{23}$$

By substituting (23), (22) is simplified to be

$$x_{t+1} = \arg\min_x z_t \cdot x + \sum_{s=1}^{t} \frac{Q_s}{2\alpha_s} \|x\|_2^2 + \Psi_t(x). \tag{24}$$

By substituting $\Psi_t(x)$ (Eq. (5)) into (24), we get

$$x_{t+1} = \arg\min_x z_t \cdot x + \sum_{g=1}^{G} \left( \lambda_1 \|x^g\|_1 + \lambda_{21} \sqrt{d_{x^g}} \|(\sum_{s=1}^{t} \frac{Q_s^g}{2\alpha_s} + \lambda_2 \mathbb{I})^{\frac{1}{2}} x^g\|_2 \right) +$$
$$\|(\sum_{s=1}^{t} \frac{Q_s}{2\alpha_s} + \lambda_2 \mathbb{I})^{\frac{1}{2}} x\|_2^2. \tag{25}$$

Since the objective of (25) is component-wise and element-wise, we can focus on the solution in one group, say $g$, and one entry, say $i$, in the $g$-th group. Let $\sum_{s=1}^{t} \frac{Q_s^g}{2\alpha_s} = \mathrm{diag}(\sigma_t^g)$ where $\sigma_t^g = (\sigma_{t,1}^g, \ldots, \sigma_{t,d_{x^g}}^g)$. The objective of (25) on $x_{t+1,i}^g$ is

$$\Omega(x_{t+1,i}^g) = z_{t,i}^g x_{t+1,i}^g + \lambda_1 |x_{t+1,i}^g| + \Phi(x_{t+1,i}^g), \tag{26}$$

where $\Phi(x_{t+1,i}^g) = \lambda_{21} \sqrt{d_{x^g}} \|(\sigma_{t,i}^g + \lambda_2)^{\frac{1}{2}} x_{t+1,i}^g\|_2 + \|(\sigma_{t,i}^g + \lambda_2)^{\frac{1}{2}} x_{t+1,i}^g\|_2^2$ is a non-negative function and $\Phi(x_{t+1,i}^g) = 0$ iff $x_{t+1,i}^g = 0$ for all $i \in \{1, \ldots, d_{x^g}\}$.

We discuss the optimal solution of (26) in three cases:

a) If $z_{t,i}^g = 0$, then $x_{t+1,i}^g = 0$.

b) If $z_{t,i}^g > 0$, then $x_{t+1,i}^g \leq 0$. Otherwise, if $x_{t+1,i}^g > 0$, we have $\Omega(-x_{t+1,i}^g) < \Omega(x_{t+1,i}^g)$, which contradicts the minimization value of $\Omega(x)$ on $x_{t+1,i}^g$.

   Next, if $z_{t,i}^g \leq \lambda_1$, then $x_{t+1,i}^g = 0$. Otherwise, if $x_{t+1,i}^g < 0$, we have $\Omega(x_{t+1,i}^g) = (z_{t,i}^g - \lambda_1) x_{t+1,i}^g + \Phi(x_{t+1}^{g,i}) > \Omega(0)$, which also contradicts the minimization value of $\Omega(x)$ on $x_{t+1,i}^g$.

   Third, $z_{t,i}^g > \lambda_1$ $(\forall i = 1, \ldots, d_{x^g})$. The objective of (26) for the $g$-th group, $\Omega(x_{t+1}^g)$, becomes

$$(z_t^g - \lambda_1 \mathbf{1}_{d_{x^g}}) \cdot x_{t+1}^g + \Phi(x_{t+1}^g).$$

c) If $z_{t,i}^g < 0$, the analysis is similar to b). We have $x_{t+1,i}^g \geq 0$. When $-z_{t,i}^g \leq \lambda_1$, $x_{t+1,i}^g = 0$. When $-z_{t,i}^g > \lambda_1$ $(\forall i = 1, \ldots, d_{x^g})$, we have

$$\Omega(x_{t+1}^g) = (z_t^g + \lambda_1 \mathbf{1}_{d_{x^g}}) \cdot x_{t+1}^g + \Phi(x_{t+1}^g).$$

From a), b), c) above, we have

$$x_{t+1}^g = \arg\min_x -s_t^g \cdot x + \Phi(x), \tag{27}$$

where the $i$-th element of $s_t^g$ is defined same as (9).

Define

$$y = (\mathrm{diag}(\sigma_t^g) + \lambda_2 \mathbb{I})^{\frac{1}{2}} x. \tag{28}$$

By substituting (28) into (27), we get

$$y_{t+1}^g = \arg\min_y -\tilde{s}_t^g \cdot y + \lambda_{21} \sqrt{d_{x^g}} \|y\|_2 + \|y\|_2^2, \tag{29}$$

where $\tilde{s}_t^g = (\text{diag}(\sigma_t^g) + \lambda_2 \mathbb{I})^{-1} s_t^g$ which is defined same as (10). This is unconstrained non-smooth optimization problem. Its optimality condition (see Rockafellar (1970), Section 27) states that $y_{t+1}^g$ is an optimal solution if and only if there exists $\xi \in \partial \|y_{t+1}^g\|_2$ such that

$$-\tilde{s}_t^g + \lambda_{21}\sqrt{d_{x^g}}\xi + 2y_{t+1}^g = 0. \tag{30}$$

The subdifferential of $\|y\|_2$ is

$$\partial\|y\|_2 = \begin{cases} \{\zeta \in \mathbb{R}^{d_{x^g}} \mid -1 \le \zeta^{(i)} \le 1, i = 1, \ldots, d_{x^g}\} & \text{if } y = 0, \\ \frac{y}{\|y\|_2} & \text{if } y \neq 0. \end{cases}$$

Similarly to the analysis of $\ell_1$-regularization, we discuss the solution of (30) in two different cases:

a) If $\|\tilde{s}_t^g\|_2 \le \lambda_{21}\sqrt{d_{x^g}}$, then $y_{t+1}^g = 0$ and $\xi = \frac{\tilde{s}_t^g}{\lambda_{21}\sqrt{d_{x^g}}} \in \partial\|0\|_2$ satisfy (30). We also show that there is no solution other than $y_{t+1}^g = 0$. Without loss of generality, we assume $y_{t+1,i}^g \neq 0$ for all $i \in \{1, \ldots, d_{x^g}\}$, then $\xi = \frac{y_{t+1}^g}{\|y_{t+1}^g\|_2}$, and

$$-\tilde{s}_t^g + \frac{\lambda_{21}\sqrt{d_{x^g}}}{\|y_{t+1}^g\|_2}y_{t+1}^g + 2y_{t+1}^g = 0. \tag{31}$$

From (31), we can derive

$$(\frac{\lambda_{21}\sqrt{d_{x^g}}}{\|y_{t+1}^g\|_2} + 2)\|y_{t+1}^g\|_2 = \|\tilde{s}_t^g\|_2.$$

Furthermore, we have

$$\|y_{t+1}^g\|_2 = \frac{1}{2}(\|\tilde{s}_t^g\|_2 - \lambda_{21}\sqrt{d_{x^g}}), \tag{32}$$

where $\|y_{t+1}^g\|_2 > 0$ and $\|\tilde{s}_t^g\|_2 - \lambda_{21}\sqrt{d_{x^g}} \le 0$ contradict each other.

b) If $\|\tilde{s}_t^g\|_2 > \lambda_{21}\sqrt{d_{x^g}}$, then from (31) and (32), we get

$$y_{t+1}^g = \frac{1}{2}(1 - \frac{\lambda_{21}\sqrt{d_{x^g}}}{\|\tilde{s}_t^g\|_2})\tilde{s}_t^g. \tag{33}$$

We replace $y_{t+1}^g$ of (33) by $x_{t+1}^g$ using (28), then we have

$$\begin{aligned} x_{t+1}^g &= (\text{diag}(\sigma_t^g) + \lambda_2\mathbb{I})^{-\frac{1}{2}}y_{t+1}^g \\ &= (2\text{diag}(\sigma_t^g) + 2\lambda_2\mathbb{I})^{-1}(1 - \frac{\lambda_{21}\sqrt{d_{x^g}}}{\|\tilde{s}_t^g\|_2})s_t^g \\ &= (\sum_{s=1}^t \frac{Q_s}{\alpha_s} + 2\lambda_2\mathbb{I})^{-1}(1 - \frac{\lambda_{21}\sqrt{d_{x^g}}}{\|\tilde{s}_t^g\|_2})s_t^g. \end{aligned} \tag{34}$$

Combine a) and b) above, we finish the proof. $\square$

## D PROOF OF THEOREM 2

*Proof.* We use the method of induction.

a) When $t = 1$, then Algorithm 1 becomes

$$Q_1 = \alpha_1(\frac{\sqrt{V_1}}{\alpha_1} - \frac{\sqrt{V_0}}{\alpha_0}) = \sqrt{V_1},$$

$$z_1 = z_0 + m_1 - \frac{Q_1}{\alpha_1}x_1 = m_1 - \frac{\sqrt{V_1}}{\alpha_1}x_1,$$

$$s_1 = -z_1 = \frac{\sqrt{V_1}}{\alpha_1}x_1 - m_1,$$

$$x_2 = (\frac{\sqrt{V_1}}{\alpha_1})^{-1}s_1 = x_1 - \alpha_1\frac{m_1}{\sqrt{V_1}},$$

which equals to Eq. (1).

b) Assume $t = T$, Eq. (35) are true.

$$z_T = m_T - \frac{\sqrt{V_T}}{\alpha_T} x_T, \quad x_{T+1} = x_T - \alpha_T \frac{m_T}{\sqrt{V_T}}. \tag{35}$$

For $t = T + 1$, we have

$$
\begin{aligned}
z_{T+1} &= z_T + m_{T+1} - \frac{Q_{T+1}}{\alpha_{T+1}} x_{T+1} \\
&= m_T - \frac{\sqrt{V_T}}{\alpha_T} x_T + m_{T+1} - \frac{Q_{T+1}}{\alpha_{T+1}} x_{T+1} \\
&= m_T - \frac{\sqrt{V_T}}{\alpha_T} (x_{T+1} + \alpha_T \frac{m_T}{\sqrt{V_T}}) + m_{T+1} - \frac{Q_{T+1}}{\alpha_{T+1}} x_{T+1} \\
&= m_{T+1} - (\frac{\sqrt{V_T}}{\alpha_T} + \frac{Q_{T+1}}{\alpha_{T+1}}) x_{T+1} = m_{T+1} - \frac{\sqrt{V_{T+1}}}{\alpha_{T+1}} x_{T+1},
\end{aligned}
$$

$$x_{T+2} = (\frac{\sqrt{V_{T+1}}}{\alpha_{T+1}})^{-1} s_{T+1} = -(\frac{\sqrt{V_{T+1}}}{\alpha_{T+1}})^{-1} z_{T+1} = x_{T+1} - \alpha_T \frac{m_{T+1}}{\sqrt{V_{T+1}}}.$$

Hence, we complete the proof. □

## E   PROOF OF THEOREM 3

*Proof.* Let

$$h_t(x) = \begin{cases} \sum_{s=1}^{t} \frac{1}{2\alpha_s} \|Q_s^{\frac{1}{2}}(x - x_s)\|_2^2 & \forall t \in [T], \\ \frac{1}{2}\|x - c\|_2^2 & t = 0. \end{cases}$$

It is easy to verify that for all $t \in [T]$, $h_t(x)$ is 1-strongly convex with respect to $\|\cdot\|_{\sqrt{V_t}/\alpha_t}$ which $\frac{\sqrt{V_t}}{\alpha_t} = \sum_{s=1}^{t} \frac{Q_s}{\alpha_s}$ , and $h_0(x)$ is 1-strongly convex with respect to $\|\cdot\|_2$.

From (7), we have

$$
\begin{aligned}
\mathcal{R}_T &= \sum_{t=1}^{T} (f_t(x_t) - f_t(x^*)) \le \sum_{t=1}^{T} \langle g_t, x_t - x^* \rangle \\
&= \sum_{t=1}^{T} \langle m_t - \gamma m_{t-1}, x_t - x^* \rangle \le \sum_{t=1}^{T} \langle m_t, x_t - x^* \rangle \\
&= \sum_{t=1}^{T} \langle m_t, x_t \rangle + \Psi_T(x^*) + h_T(x^*) + (\sum_{t=1}^{T} \langle -m_t, x^* \rangle - \Psi_T(x^*) - h_T(x^*)) \\
&\le \sum_{t=1}^{T} \langle m_t, x_t \rangle + \Psi_T(x^*) + h_T(x^*) + \sup_{x \in \mathcal{Q}} \{\langle -m_{1:T}, x \rangle - \Psi_T(x) - h_T(x)\},
\end{aligned}
\tag{36}
$$

where in the first and second inequality above, we use the convexity of $f_t(x)$ and the condition (12) respectively.

We define $h_t^*(u)$ to be the conjugate dual of $\Psi_t(x) + h_t(x)$:

$$h_t^*(u) = \sup_{x \in \mathcal{Q}} \{\langle u, x \rangle - \Psi_t(x) - h_t(x)\}, \quad t \ge 0,$$

where $\Psi_0(x) = 0$. Since $h_t(x)$ is 1-strongly convex with respect to the norm $\|\cdot\|_{h_t}$, the function $h_t^*$ has 1-Lipschitz continuous gradients with respect to $\|\cdot\|_{h_t^*}$ (see, Nesterov (2005), Theorem 1):

$$\|\nabla h_t^*(u_1) - \nabla h_t^*(u_2)\|_{h_t} \le \|u_1 - u_2\|_{h_t^*}, \tag{37}$$

and

$$\nabla h_t^*(u) = \arg\min_{x \in \mathcal{Q}} \{-\langle u, x \rangle + \Psi_t(x) + h_t(x)\}. \tag{38}$$

As a trivial corollary of (37), we have the following inequality:

$$h_t^*(u + \delta) \le h_t^*(u) + \langle \nabla h_t^*(u), \delta \rangle + \frac{1}{2}\|\delta\|_{h_t^*}^2. \tag{39}$$

Since $h_{t+1}(x) \ge h_t(x)$ and $\Psi_{t+1}(x) \ge \Psi_t(x)$, from (38), (39), (6), we have

$$
\begin{aligned}
h_T^*(-m_{1:T}) &\le h_{T-1}^*(-m_{1:T}) \\
&\le h_{T-1}^*(-m_{1:T-1}) - \langle \nabla h_{T-1}^*(-m_{1:T-1}), m_T \rangle + \frac{1}{2}\|m_T\|_{h_{T-1}^*}^2 \\
&\le h_{T-2}^*(-m_{1:T-1}) - \langle x_T, m_T \rangle + \frac{1}{2}\|m_T\|_{h_{T-1}^*}^2 \\
&\le h_0^*(0) - \langle \nabla h_0^*(0), m_1 \rangle - \sum_{t=2}^{T} \langle x_t, m_t \rangle + \frac{1}{2}\sum_{t=2}^{T}\|m_t\|_{h_{t-1}^*}^2 \\
&= -\sum_{t=1}^{T} \langle x_t, m_t \rangle + \frac{1}{2}\sum_{t=1}^{T}\|m_t\|_{h_{t-1}^*}^2.
\end{aligned}
\tag{40}
$$

where the last equality above follows from $h_0^*(0) = 0$ and (11) which deduces $x_1 = \nabla h_0^*(0)$.

By substituting (40), (36) becomes

$$
\begin{aligned}
\mathcal{R}_T &\le \sum_{t=1}^{T} \langle m_t, x_t \rangle + \Psi_T(x^*) + h_T(x^*) + h_T^*(-m_{1:T}) \\
&\le \Psi_T(x^*) + h_T(x^*) + \frac{1}{2}\sum_{t=1}^{T}\|m_t\|_{h_{t-1}^*}^2.
\end{aligned}
\tag{41}
$$

$\square$

# F   ADDITIONAL PROOFS

## F.1   PROOF OF LEMMA 1

*Proof.* Let $V_t = \mathrm{diag}(\sigma_t)$ where $\sigma_t$ is the vector of the diagonal elements of $V_t$. For $i$-th entry of $\sigma_t$, by substituting (13) into (15), we have

$$
\begin{aligned}
\sigma_{t,i} = g_{t,i}^2 + \eta\sigma_{t-1,i} &= (m_{t,i} - \gamma m_{t-1,i})^2 + \eta g_{t-1,i}^2 + \eta^2 \sigma_{t-2,i} \\
&= \sum_{s=1}^{t} \eta^{t-s}(m_{s,i} - \gamma m_{s-1,i})^2 \ge \sum_{s=1}^{t} \eta^{t-s}(1-\gamma)(m_{s,i}^2 - \gamma m_{s-1,i}^2) \\
&= (1-\gamma)\Big(m_{t,i}^2 + (\eta - \gamma)\sum_{s=1}^{t-1} \eta^{t-s-1} m_{s,i}^2\Big).
\end{aligned}
\tag{42}
$$

Next, we will discuss the value of $\eta$ in two cases.

a) $\eta = 1$. From (42), we have

$$\sigma_{t,i} \ge (1-\gamma)\Big(m_{t,i}^2 + (1-\gamma)\sum_{s=1}^{t-1} m_{s,i}^2\Big) > (1-\gamma)^2 \sum_{s=1}^{t} m_{s,i}^2 \ge (1-\nu)^2 \sum_{s=1}^{t} m_{s,i}^2. \tag{43}$$

Recalling the definition of $M_{t,i}$ in Section 1.5, from (43), we have

$$\sum_{t=1}^{T} \frac{m_{t,i}^2}{\sqrt{\sigma_{t,i}}} < \frac{1}{1-\nu}\sum_{t=1}^{T} \frac{m_{t,i}^2}{\|M_{t,i}\|_2} \le \frac{2}{1-\nu}\|M_{T,i}\|_2,$$

where the last inequality above follows from Appendix C of Duchi et al. (2011). Therefore, we get

$$\sum_{t=1}^{T} \|m_t\|_{(\frac{\sqrt{V_t}}{\alpha_t})^{-1}}^2 = \alpha \sum_{t=1}^{T}\sum_{i=1}^{d} \frac{m_{t,i}^2}{\sqrt{\sigma_{t,i}}} < \frac{2\alpha}{1-\nu}\sum_{i=1}^{d} \|M_{T,i}\|_2. \tag{44}$$

b) $\eta < 1$. We assume $\eta \geq \gamma$ and $\kappa V_t \succeq V_{t-1}$ where $\kappa < 1$, then we have

$$\sum_{s=1}^{t} \kappa^{t-s} \sigma_{t,i} \geq \sum_{s=1}^{t} \sigma_{s,i} \geq (1-\gamma) \sum_{s=1}^{t} m_{s,i}^2.$$

Hence, we get

$$\sigma_{t,i} \geq \frac{1-\kappa}{1-\kappa^t}(1-\gamma)\sum_{s=1}^{t} m_{s,i}^2 > (1-\kappa)(1-\gamma)\sum_{s=1}^{t} m_{s,i}^2 \geq (1-\nu)^2 \sum_{s=1}^{t} m_{s,i}^2, \quad (45)$$

which deduces the same conclusion (44) of a).

Combine a) and b), we complete the proof. □

### F.2 PROOF OF COROLLARY 1

*Proof.* From the definition of $m_t$ (13), $V_t$ (15), we have

$$|m_{t,i}| = |\sum_{s=1}^{t} \gamma^{t-s} g_{s,i}| \leq \frac{1-\gamma^t}{1-\gamma}G < \frac{G}{1-\gamma} \leq \frac{G}{1-\nu},$$

$$|\sigma_{t,i}| = |\sum_{s=1}^{t} \eta^{t-s} g_{s,i}^2| \leq tG^2.$$

Hence, we have

$$\Psi_T(x^*) \leq \lambda_1 dD_1 + \lambda_{21} dD_1\left(\frac{\sqrt{T}G}{2\alpha} + \lambda_2\right)^{\frac{1}{2}} + \lambda_2 dD_1^2, \quad (46)$$

$$h_T(x^*) \leq \frac{dD_2^2 G}{2\alpha}\sqrt{T}, \quad (47)$$

$$\frac{1}{2}\sum_{t=1}^{T} \|m_t\|_{h_{t-1}^*}^2 < \frac{\alpha}{1-\nu}\sum_{i=1}^{d} \frac{\sqrt{T}G}{1-\nu} = \frac{d\alpha G}{(1-\nu)^2}\sqrt{T}. \quad (48)$$

Combining (46), (47), (48), we complete the proof. □

## G ADDITIONAL EMPIRICAL RESULTS

Table 7: AUC of MLP for different embedding dimensions and sparsity (feature rate) in parentheses. The best results are bolded.

| Embedding Dimension | GROUP ADAM |
|---|---|
| 4 | 0.7462 (0.123) |
| 8 | 0.7471 (0.056) |
| 16 | **0.7486**[*](0.018) |
| 32 | 0.7480 (**0.006**) |

[*] It is significantly better than embedding dimensions of 4, 8 but has no difference in 95% confidence level of the embedding dimension of 32.

Table 8: Sparsity (feature rate) of MLP for different values of $\lambda_{21}$, $\lambda_1$ and AUC in parentheses.

| $\lambda_{21}$ \ $\lambda_1$ | 0 | 1e-4 | 5e-4 | 1e-3 | 5e-3 | 1e-2 |
|---|---|---|---|---|---|---|
| 0 | - | 0.987 (0.7486) | 0.927 (0.7482) | 0.866 (0.7485) | 0.470 (0.7481) | 0.214 (0.7475) |
| 1e-4 | 0.971 (0.7477) | - | 0.902 (0.7486) | 0.839 (0.7484) | 0.458 (0.7480) | 0.212 (0.7483) |
| 5e-4 | 0.867 (0.7490) | 0.829 (0.7485) | - | 0.682 (0.7483) | 0.344 (0.7485) | 0.169 (0.7480) |
| 1e-3 | 0.702 (0.7477) | 0.684 (0.7477) | 0.612 (0.7480) | - | 0.274 (0.7479) | 0.134 (0.7478) |
| 5e-3 | 0.136 (0.7485) | 0.138 (0.7484) | 0.120 (0.7482) | 0.106 (0.7482) | - | 0.035 (0.7483) |
| 1e-2 | 0.033 (0.7481) | 0.037 (0.7480) | 0.033 (0.7481) | 0.029 (0.7485) | 0.018 (0.7486) | - |

