# OpenReview forum: "Adaptive Optimizers with Sparse Group Lasso"
_ICLR.cc/2021/Conference — Reject_

### Official Review · AnonReviewer3 · 2020-10-26
**Not sure about the novelty**

**Rating:** 3
**Confidence:** 4

**Review:**

This work studies the adaptive proximal gradient descent method, and specifically studies the group sparsity. To encourage the group sparsity, a regularizer which is a combination of $\ell_1$ norm, block $\ell_1$ norm and $\ell_2$ norm square is used. This paper gives the update rule of the proximal gradient with the specific regularizer. After proposing the update rule, the paper analyzes the convergence and regret guarantee of the algorithm.

However, I'm not sure if the contribution is enough for the conference, as it is known that the block $\ell_1$ norm can encourage the block sparsity, and the computation of proximal gradient is fairly standard and straightforward. The convergence of proximal gradient method is not too different from gradient method as well.

I think it can be more interesting if the work can focus on the statistical property of the regularizer $\Psi$. As suggested in Oymak et al, summing up a few regularization terms might not actually benefit with getting the structure. Analyzing whether the solution of the objective is group sparse, and whether one can find the group sparse parameter with less data than solving with unregularized least squares, is more interesting and less exploited.

================================= Update ===================================

I slightly raise my rating, as everything is correct and well organized. However I'm still not sure if this is enough contribution or just incremental compared to the existing computation of proximal gd. I'd leave it to other reviewers.

---

> ### Author Response · Authors · 2020-11-25
> **Authors' Response**
>
> Thanks for your comments. We will explain your concerns about novelty.
>
> First, our contribution is to propose a general framework that adds the regularizers to a family of adaptive optimizers in deep learning, and create a new class of optimizers. The new optimizers maintain the characteristics of the original optimizers, and introduce additional functionality of producing sparsity. The new optimizers are suitable to the application domains which have redundant or noise features, whose gain is from deleting the redundant features and letting the model more generalized. The new optimizers are also suitable to the application whose serving resource is limited, e.g., embedded devices with limited memory space, and sparsity might be relatively more important.
>
> Second, either the derivation of the closed-form solution or the proof of convergence, it leverages many mathematical skills which are totally different from gradient methods. In practical algorithms, it is quite different from the original optimizers though we have proved they are equivalent when the regularization terms of new optimizers vanish. It is also quite different from FTRL, e.g., the computation of the linear term $z_t$ which depends on the specific optimizer, and $\ell_{21}$-regularization.
>
> Third, as we clarify in the revision submitted, each group of DNN models is defined as the set of outgoing weights from a unit which can be an input feature, or a hidden neuron, or a bias unit. In our experiments, each embedding vector is considered as a group. Therefore, the number of features is just the number of groups. Using $\ell_{21}$-regularization is actually to exploit the data structure.

---

### Official Review · AnonReviewer2 · 2020-10-29
**The paper proposes a novel framework to add sparsity regularizers to adaptive optimizers in deep learning such as Adam, Adagrad... A generic algorithm is proposed and the theoretical convergence and regret guarantees are provided. Empirical evaluations on ad click logs datasets are provided to show the effectiveness of the framework.**

**Rating:** 5
**Confidence:** 3

**Review:**

Summary:

The paper proposes a novel framework to add sparsity regularizers to adaptive optimizers in deep learning such as Adam, Adagrad... A generic algorithm is proposed and the theoretical convergence and regret guarantees are provided. Empirical evaluations on ad click logs datasets are provided to show the effectiveness of the framework



Pros:

- The idea of using sparse regularization (sparse group lasso) in the general update formulation of adaptive optimizers is interesting. It provides a unified way to derive practical optimization algorithm of deep networks parameters under sparsity constraints. Re-assuringly the proposed algorithm reduces to the conventional adaptive optimizers when the regularization parameters are set to zeros.
- The proposed formulation and algorithm is theoretically analyzed. Convergence guarantees, in terms of bound on the regret, are established using results from stochastic convex optimization and based on primal-dual analysis. The guarantees ensure a convergence rate of $\sqrt{1/T}$. These results are interesting per se although they are not exploited in practice to control the algorithm convergence.
- Evaluation of the new optimizers (including the sparse group lasso regularizer) to train deep networks on three real-world ad click datasets highlight the fact that the new optimizers achieve highly sparse layer weights with competitive or better accuracy performances compared to the classical un-regularized version of the optimizers (plain Adam, Adagrad…). The considered datasets are characterized by categorical features which are pre-processed using one-hot encoding and leads to sparse high-dimension inputs for the deep networks. The reported results shows the practical effectiveness of the method for ad click prediction.

Concerns:
- A concern about the paper is the lack of justification of the proposed regularization scheme. The rationale behind the regularization term, Equation (5), is insufficiently discussed and explained. For instance,  it is unclear why the “second order” matrix $Q_s^g/\alpha_s + \lambda_2 \mathbf{I}$ is used in the group lasso term. Also, it is unclear why the same regularization parameter $\lambda_2$ is used for the group lasso penalty and the $\ell_2$-norm penalty.
- The new optimizers GROUP ADAGRAD, GROUP ADAM… involve three additional hyper-parameters $\lambda_1$, $\lambda_2$, $\lambda_{12}$  which tuning may be tedious compared with their un-regularized counterparts.
-The introduction of the term $(\sum_s Q_s^g/\alpha_s + \lambda_2 \mathbf{I})^{1/2}$ in the group lasso regularization is justified by the fact that a close form solution (Equation 8) is attainable in such setting. However in Algorithm 1 this exact closed form is not used as $\tilde{s}$ is replaced by $s_t$ in Equation (8). Here again this choice is not well justified. What does it entail to use $\tilde{s}$?
- The empirical evaluations are restricted to ad click datasets with highly sparse input vectors. Does it mean that the proposed group optimizers are only suited in that setting? If so, which other application domains may benefit from the group sparse optimizers?  Nevertheless, a deeper analysis of the proposed method on other application domains such computer vision (Adam optimizer is customary used in the latter domain) would be nice.
- For the reproducibility sake, the paper should mention how the parameter groups are selected beforehand. A claim of the paper is the enhanced sparsity achieved by the new optimizers GROUP ADAGRAD, GROUP ADAM compared to Adam, Adagrad even when the latter are used to optimize the fitting term with a group lasso regularization. The paper should make explicit how the training with Adam or Adagrad was implemented in that setting. Does the training use a proximal-based at each iteration?
- To improve the readability of the paper, the most prominent empirical results should be moved in the main paper. All the experimental results can not be deferred to the appendix as this leads to unpleasant to read Section 4.


Other comments
- In Lemma 1, the setting of $\eta$ in assumptions 1 and 2 is confusing.
- In Equation (16) what is $d$?
- In Lemma 2, there are some trailing dot symbols in the definition of the dual norms. They should be removed.

After rebuttal:
- I  have read the response of the authors.
- Some concerns are adressed: for instance some empirical results are moved to the main paper,  the tuning of the hyper-parameters is discussed and details about the groups of variables are provided.
- Nevertheless it is still unclear why $s_t$ instead of $\tilde{s}$ is  used in  algorithm 1 and what might be the performances of the deep models  trained using $\tilde{s}$.
- The baseline model is now moved to Adam with weight pruning. One concern of the review is to detail  how Adam, Adagrad are used to optimize the fitting term with a group lasso regularization. This point was skipped in the new version, hence it's not easy to assess the effectiveness of the Group Adam, Group Adagrad methods.

---

> ### Author Response · Authors · 2020-11-25
> **Authors' Response**
>
> Thanks for your constructive and valuable comments. We will explain your concerns point by point.
> 1. In our revision, we replace $\sum_{s=1}^t \frac{Q_s^g}{2\alpha_s} + \lambda_{2}\mathbb{I}$ with $A_t$ which can be arbitrary positive matrix satisfying $A_{t+1}\succeq A_t$, e.g., $A_t = \mathbb{I}$, in Eq. (5), and the convergence is still established. The only reason we add $\sum_{s=1}^t \frac{Q_s^g}{2\alpha_s} + \lambda_{2}\mathbb{I}$ to group lasso penalty is to solve the closed-form directly since the formulas (28)-(29) of the proof of Theorem 1. But in actual algorithm, we eliminate the influence of the term $\sum_{s=1}^t \frac{Q_s^g}{2\alpha_s} + \lambda_{2}\mathbb{I}$ when do the $\ell_{21}$-regularization.
> 2. We have supplemented the practical experiences of how to tune $\lambda_{1},\ \lambda_{2},\ \lambda_{21}$ in the revision submitted. Since we show $\ell_{21}$ is much more effective than $\ell_{1}$ in producing sparsity in Section 4.4, we can only tune $\lambda_{21}$ and use default settings for $\lambda_{2}$ and $\lambda_{1}$, i.e., $\lambda_2=1\text{e-5}$ and $\lambda_1=0$, if we just want to produce sparsity. If we need to further improve the performance of the model, we need to tune $\lambda_{21}$ and $\lambda_{1}$, $\lambda_2=1\text{e-5}$ is always as a good default setting.
> 3. Followed by Answer 1 above, we realize the part $\sum_{s=1}^t \frac{Q_s^g}{2\alpha_s} + \lambda_{2}\mathbb{I}$ of $\tilde{s_t}$ is not necessary. Just as explained in our paper, our purpose is to let every entry of the group have the same effect of $\ell_{21}$-regularization. Therefore, we replace $\tilde{s_t}$ with $s_t$. The convergence of this modified version is unclear since we can't find the explicit formula of $A_t$ to derive the modified version so far. However , it works well in practice, just like Adam which has no convergence guarantee. The similar example is FTRL. In the theoretical convergence analysis, the update formula is $x_{t+1}=\arg\min_{x} g_{1:t} + t\lambda\\|x\\|_1 + \frac{1}{2}\sum_\{s=1\}^t\\|Q_s^{1/2}(x-0)\\|_2^2$ (https://arxiv.org/abs/1009.3240). But in practical algorithm, $t\lambda\\|x\\|_1$ is adjusted to $\lambda\\|x\\|_1$ (https://static.googleusercontent.com/media/research.google.com/en//pubs/archive/41159.pdf).
> 4. The new optimizers are suitable to the application domains which have redundant or noise features. In fact, the gain of the new optimizers is from deleting the redundant features and letting the model more generalized. The new optimizers are also suitable to the applications whose serving resource is limited, e.g., the embedded device with expensive memory, and sparsity might be relatively more important. Since now we focus on the field of large scale online learning, we are not sure computer vision (every feature seems to be meaningful) or natural language processing can get great profit from new optimizers. It would be interesting in the future to evaluate our optimizers on these applications.
> 5. We have supplemented the explicit definition of group in the revision. In DNN models, each group is defined as the set of outgoing weights from a unit which can be an input feature, or a hidden neuron, or a bias unit. In our experiments, each embedding vector is considered as a group. Therefore, the number of features is just the number of the groups. Since some reviewer doubted the fairness of the first experiment of Section 4.2, we change in the first experiment to use the magnitude pruning method as the baseline, and compare the new optimizer and original optimizer on the same sparsity level. Therefore, the original optimizers Adam and Adagrad were implemented as same as community. The only difference is in the first experiment of Section 4.2 the original optimizer Adam needs post-processing procedure which uses the magnitude pruning method.
> 6. We have moved the most prominent empirical results to the main text in the revision submitted.
>
> Answers to other comments:
> 1. The conditions (1) and (2) of Lemma 1 come from the formula (42) of the proof of Lemma 1. When $\eta = 1$, it will reduce to (45) directly, and when $\eta < 1$, it needs additional constraints.
> 2. $d$ is the dimension of $x_t$. We have supplemented to Lemma 1 in the revision submitted.
> 3. The same symbol also appeared in Section 1.1 of  https://www.jmlr.org/papers/volume12/duchi11a/duchi11a.pdf, i.e., $\\|\cdot\\|_A = \sqrt{\left<\cdot, A\cdot\right>}$, so we think our symbols are right.

---

### Official Review · AnonReviewer4 · 2020-10-30
**The proposed regularizers for adaptive optimizer for neural network has no convergence guarantee for nonconvex case. The experiments are weak and the results are not convincing.**

**Rating:** 4
**Confidence:** 4

**Review:**

The contribution is to introduce regularizers to adaptive optimizer for neural network optimization. Convergence guarantee is provided for the convex case. The authors claim that the proposed method achieves competitive performance but with more sparse solutions.

Pros:
1. The introduced regularizer is general. It extends several existed adaptive optimizers. The extension is simple and the pursuit of sparse solution is reasonable and useful.
2. The authors provide the convergence guarantee.

Cons:
1. This work focuses the study on the solver for neural network optimization. However, the convergence guarantee is only for convex case. In the convex case, the authors fail to show that it is better than existed solvers, e.g., the convergence bound is tighter (showing the optimal convergence rate is not enough). Without the convergence guarantee for the nonconvex case, the results are weak.
2. The main contribution is the regularizers which can be added in existed adaptive optimizer. Though the motivation is to find more sparse solutions, the formulation in (5) is a bit complicated without detailed explanation, e.g., the weights for the groups are so complicated. The regularizers are not practical since there have many parameters, e.g., $\lambda_1$, $\lambda_2$, $\lambda_{21}$, the size of groups, the way of the group partition. The authors do not provide any suggestions on their choices. It is difficult to tune so many parameters in practice.
3. The experimental results are not very convincing.
(1) In the experiments, the authors evaluate the performance of optimizers mainly based on the learning performance and sparsity. However, the learning performance may depend on many different factors.  Showing the convergence of the objective function values (or other related functions) in each iterations is much more important.
(2) The comparison in experiments may not be fair.  In the first experiment in section 4.2, the authors compare the proposed method with the traditional method by adding the regularization term to the loss. In this setting, both the model and the solver
are different. It is hard to say that the proposed regularizer makes the results better. The roles of the regularization terms and the values of $\ell_{21}$ are quite different in two compared models. It may not be fair to compare them based on the same value of $\ell_{21}$. The above issues also appear in other experiments.
(3) From the results in Table 8, the authors claim that $\ell_{21}$-regularization is much more effective than `$\ell_1$-regularization in producing sparsity. This is not convincing since the best value of $\ell_{21}$ and $\ell_1$ may be quite different.

Other questions:
1. How to choose the parameters $\lambda_1$, $\lambda_2$ and $\lambda_{21}$ in experiments (though the authors show them in Table 3 for the experiments)? How to do the group partition?
2. Section 2.1, the authors slightly modify (8) by setting $\hat{s}_t=s_t$. Will this modification affect the algorithm and its convergence analysis? If the convergence analysis is for the modified version, it is not necessary to introduce (8) for avoiding confusion.
3. The parameters $\ell_1$, $\ell_2$ and $\ell_{21}$ in the experiments should be  $\lambda_1$, $\lambda_2$ and $\lambda_{21}$ in (5).

---

> ### Author Response · Authors · 2020-11-25
> **Authors' Response**
>
> Thanks for your constructive and valuable comments. We will explain your concerns point by point.
> 1. Since we need to deal with a family of adaptive optimizers, it is not easy to get tighter bound. We admit that it is unclear about the convergence in non-convex settings and we plan to investigate this work in the future.
> 2. We have added more explanations in the revision. In DNN models, each group is defined as the set of outgoing weights from a unit which can be an input feature, or a hidden neuron, or a bias unit. In our experiments, each embedding vector is considered as a group, and a visual comparison between $\ell_1$, $\ell_{21}$ and mixed regularization effect is given in Fig. 2 of https://arxiv.org/pdf/1607.00485.pdf. Since we show $\ell_{21}$ is much more effective than $\ell_{1}$ in producing sparsity in Section 4.4, we can only tune $\lambda_{21}$ and use default settings for $\lambda_{2}$ and $\lambda_{1}$, i.e., $\lambda_2=1\text{e-5}$ and $\lambda_1=0$, if we just want to produce sparsity. If we need to further improve the performance of the model, we need to tune $\lambda_{21}$ and $\lambda_{1}$, $\lambda_2=1\text{e-5}$ is always as a good default setting. In Eq. (5), we replace $\sum_{s=1}^t \frac{Q_s^g}{2\alpha_s} + \lambda_{2}\mathbb{I}$ with $A_t$ which can be arbitrary positive matrix satisfying $A_{t+1}\succeq A_t$, e.g., $A_t = \mathbb{I}$, to simplify the formula and make it more generalized.
> 3. Partially agree. (1) In recommendation field, there are many redundant or noise features. It is possible that the model has the lower training loss just because of overfitting the noise data. In fact, the gain of the new optimizers are from deleting the redundant features and letting the model more generalized. From the experiences in industry, we mainly focus on the AUC of test set. It is one of the most relative metric with the actual effect in real world. (2) It is indeed ambiguous that adding the same value of the regularization terms to the loss in the first experiment of Section 4.2. Therefore, in the revision, we change in the first experiment to use the magnitude pruning method as the baseline, and compare the new optimizer and original optimizer on the same sparsity level. Other experiments have no such issues since the only difference is the new optimizers and the original optimizers, and our main point concerns the sparsity number. (3) We have supplemented more comprehensive results to Table 8 in the revision. It is shown that the sparsity of the (i, j) (i > j) element of Table 8 is consistently lower than the (j, i) element, therefore it is proven that $\ell_{21}$ is much more effective than $\ell_{1}$ in producing sparsity.
>
> Answers to other questions:
> 1. Please refer to the aforementioned Answer 2 according to the choice of $\lambda_1,\ \lambda_2,\ \lambda_{21}$. It doesn't need to do group partition since every embedding vector is a group.
> 2. In the modified paper, we replace $\sum_{s=1}^t \frac{Q_s^g}{2\alpha_s} + \lambda_{2}\mathbb{I}$ with $A_t$ which can be arbitrary positive matrix satisfying $A_{t+1}\succeq A_t$ in Eq. (5), and the convergence is still established. Then we let $A_t=\sum_{s=1}^t \frac{Q_s^g}{2\alpha_s} + \lambda_{2}\mathbb{I}$ for solving the closed-form directly. But we realize the part $\sum_{s=1}^t \frac{Q_s^g}{2\alpha_s} + \lambda_{2}\mathbb{I}$ of $\tilde{s_t}$ is not necessary. Just as explained in our paper, our purpose is to let every entry of the group have the same effect of $\ell_{21}$-regularization. The convergence of this modified version is unclear since we can't find the explicit formula of $A_t$ to derive the modified version so far. However, it works well in practice, just like Adam which has no convergence guarantee. One similar example is FTRL. In the theoretical convergence analysis, the update formula is $x_{t+1}=\arg\min_{x} g_{1:t} + t\lambda\\|x\\|_1 + \frac{1}{2} \sum^{t}_\{s=1\} \\|Q_s^{1/2}(x-0)\\|_2^2$ (https://arxiv.org/abs/1009.3240). But in practical algorithm, $t\lambda\\|x\\|_1$ is adjusted to $\lambda\\|x\\|_1$ (https://static.googleusercontent.com/media/research.google.com/en//pubs/archive/41159.pdf).
> 3. We have changed $\ell_{1},\ \ell_{2},\ \ell_{21}$ to $\lambda_{1},\ \lambda_{2},\ \lambda_{21}$ in the experiments in the revision submitted.

---

### Official Review · AnonReviewer5 · 2020-11-07
**Introducing group sparsity for AdaGrad/Adam via standard framework**

**Rating:** 5
**Confidence:** 3

**Review:**

Authors propose addition of group sparsity regularizer into the FTRL framework, and derive update rules of AdaGrad/Adam. They demonstrate the effectiveness by inducing sparsity on several models used in the benchmarks.

Reason to Score: Weaker experimentation, lack of standard baselines -- including them can improve the paper.

I have listed my concerns below and hopefully authors can address them during the rebuttal period.

Questions/Comments:

1. Could authors contrast their work with algorithm presented in:
https://static.googleusercontent.com/media/research.google.com/en//pubs/archive/41159.pdf
which includes an implementation in: https://www.tensorflow.org/api_docs/python/tf/keras/optimizers/Ftrl

Is the contribution an extension by using group sparsity?

2. Missing word in sentence in abstract ("not only can the")
... the loss functions, not only can the dimensions of features be effectively and efficiently reduced ...

3. Incorrect citation

Any regret minimizing algorithm can be converted to a stochastic optimization algorithm with convergence rate O(RT /T) using an online-to-batch conversion technique

Please cite:
N. Littlestone. From On-Line to Batch Learning. In Proceedings of the 2nd Workshop on Computational Learning Theory, p. 269-284, 1989.

4. A major concern was on experiment sections. Authors do not mention what type of groups were used clearly, which made it hard to judge the results.

I also suggest authors include several baselines comparing with existing work:
a) block l1 (l2 of the norm of the group) as penalty to the objective
b) standard magnitude pruning. https://arxiv.org/abs/1902.09574

== Update: Nov 30 2020 ==
Thanks for the authors for the reply.  Thank you for running those experiments.

I had a few more clarifications needed from authors. (a) Magnitude pruning typically invovles a fine tuning phase after removing the weights, was this carried out? For eg: Fig 1. a behavior was why I asked this question (b) I would recommend authors to add error bars Table 2. has results that are quite close between the methods.

I raised my score but still below accept due to the above reservations.

---

> ### Author Response · Authors · 2020-11-25
> **Authors' Response**
>
> Thanks for your constructive and valuable comments. We will explain your concerns point by point.
> 1. Just like Section 2.2 of the paper said, when $\lambda_{21}=0$, Group AdaGrad is equivalent to FTRL. Therefore, FTRL is a special case of our optimizer family. In contrast to FTRL, our contribution is not an extension  of FTRL by using group sparsity which can only derive Group AdaGrad, but to propose a general framework that adds the regularizers to a family of adaptive optimizers in deep learning, and create a new class of optimizers. The new optimizers maintain the characteristics of the original optimizers, but introduce additional functionality of producing sparsity.
> 2. We have rephrased the sentence in the revision submitted.
> 3. We have corrected the citation in the revision submitted.
> 4. In DNN models, each group is defined as the set of outgoing weights from a unit which can be an input feature, or a hidden neuron, or a bias unit. In our experiments, each embedding vector is considered as a group, and a visual comparison between $\ell_1$, $\ell_{21}$ and mixed regularization effect is given in Fig. 2 of https://arxiv.org/pdf/1607.00485.pdf. We have added these explanations in the introduction and experiments of the revision submitted. \
> The advice of using the standard magnitude pruning as the baseline is especially useful. In our revision, we change to use the magnitude pruning method as the baseline, and compare the new optimizer and the original optimizer on the same sparsity level. We only conduct the experiments of $\ell_{21}$, since $\ell_{21}$ is much more effective than $\ell_{1}$ in producing sparsity as discussed in Section 4.4, and $\ell_{1}$ is an auxiliary parameter for improving the performance of the models to some degree.

---

### Decision · Program_Chairs · 2021-01-07
**Final Decision**

**Decision:**

Reject

**Comment:**

The paper presents a generic way to add group sparsity based regularizers to a family of adaptive optimizers leading to generalizations of many popular optimizers ADAM, ADAGRAD etc  to their group versions. Overall the reviewers appreciated the algorithmic contribution and its genericness in terms of application to most known adaptive optimizers. While the paper's revision during the rebuttal phase satisfied some reviewer concerns regarding the experimental baselines and the precise experimental methodology, reviewers continued to have concerns regarding the experiments performed - the potential lack of fine tuning post pruning, the use of s_t tilde as opposed to s_t in the practical algorithms amongst others listed in the review. Overall, the reviewers deemed the theoretical contribution of the paper not significant enough in terms of novelty and the decision hinged on the efficacy of the experimental evaluation - the lingering concerns for which led to the decision.